# Method-specific beliefs and subsequent contraceptive method choice: Results from a longitudinal study in urban and rural Kenya

**George Odwe** [1]*, **Yohannes Dibaba Wado**[2], **Francis Obare**[1], **Kazuyo Machiyama**[3], **John Cleland**[3]

**1** Population Council, Nairobi, Kenya, **2** African Population and Health Research Center, Nairobi, Kenya, **3** Faculty of Epidemiology and Population Health, London School of Hygiene and Tropical Medicine, London, United Kingdom

* godwe@popcouncil.org

## Abstract

### Introduction

Evidence from sub-Saharan Africa, including Kenya, shows that negative beliefs about contraceptive methods are associated with non-use. However, little is known about the relationship between contraceptive beliefs and subsequent method choice.

### Methodology

We used data from a two-year longitudinal survey of married women aged 15–39 years at enrollment from one urban site (Nairobi) and one rural site (Homa Bay) in Kenya. Analysis entails descriptive statistics and estimation of a conditional logit analysis to examine associations between method-specific beliefs and choice of injectables, implants or pills among women who were not using any method or were pregnant at baseline (round 1) but adopted these methods at 12-month follow-up (Nairobi, n = 221; Homa Bay n = 197).

### Results

Beliefs about pills, injectables and implants among non-users were generally negative. With the partial exception of the pill in Nairobi, the majority thought that each method was likely to cause serious health problems, unpleasant side effects, menstrual disruption, and would be unsafe for long-term use. In both sites, satisfied past use of a method and the perception that a method is easy to use had a major influence on method choice. Concerns about menstrual disruption and safety for long-term use were unimportant in both sites. There were some marked differences between the two sites. Beliefs about long-term fertility impairment and perceived husband approval had strong influences on choice of injectables, implants or pills in the urban site but not in the rural site.

### Conclusion

The relative importance of beliefs, some erroneous, in predisposing women to choose one method over another appears to be conditioned by the social context. There is need for

**Data Availability Statement:** There are institutional legal restrictions on sharing de-identified data sets. Data used in this paper are available on reasonable request. To access the data, request may be sent to

Population Council, Dataverse, email: publications@popcouncil.org for information on data access.

**Funding:** The study that provided data for this paper was funded by the United Kingdom's Department for International Development through the Strengthening Evidence for Programming on Unintended Pregnancy (STEP UP) Research Programme Consortium. The funders had no role in study design, data collection and analysis, decision to publish, or preparation of the manuscript.

**Competing interests:** The authors have declared that no competing interests exist.

**Abbreviations:** IUDs, Intrauterine Devices; NUHDSS, Nairobi Urban Health and Demographic Surveillance System; OCs, Oral Contraceptives; STEP UP, Strengthening Evidence for Programming on Unintended Pregnancy.

family planning counseling programmes to pay attention to erroneous beliefs and misconceptions about contraceptives.

## Introduction

A wide variety of contraceptive methods exist from which couples can choose to achieve their reproductive goals, ranging from short-acting methods such as condoms, oral contraceptives (OCs) and injectables to long-acting reversible contraception and permanent methods which do not require user's compliance, such as implants, intrauterine contraceptive devices, and voluntary surgical contraception [1, 2]. In addition, there are traditional but less effective methods such as rhythm and withdrawal. Over the past 60 years, a substantial literature, much of it qualitative, on the relationship between method-specific beliefs and choice of a method to use has accumulated and several reviews have been published [3–5]. Much of the literature for sub-Saharan Africa (SSA) focuses on negative beliefs on contraception in general or, less commonly, on specific methods. These are often labelled as myths, misperceptions, misinformation or barriers to use. Barriers to use that have been widely documented in many SSA settings, including for Kenya, include fear of side effects, damage to health, menstrual disruption, and long-term infertility [6–8].

The influence of a wide range of factors on method choice has been documented, including knowledge, availability, affordability, counselling by providers, socio-demographic characteristics and fertility preferences of women, and the views of partners and social networks. In Kenya, past research has examined socio-demographic correlates of, and trends in, method-specific use [9–11] and the influence of providers, social networks and partners [12–16]. However, variation between methods in beliefs has been little studied.

A major strand of the extensive literature on the relationship between beliefs and use, much from the USA, concerns the contraceptive attributes that women deem most important when deciding which method to adopt [17–20]. While in sub-Saharan Africa, including evidence from Kenya [8], it is established that negative views about modern contraceptive methods, in general, are associated with non-use, little is known about the relationship between method-specific beliefs and method choice. For instance, what is the relative influence of beliefs about menstrual disruption and about long-term fertility impairment in predisposing women to choose one method over an alternative? In this paper, we address this evidence-gap, building on earlier research from the same project [6]. Specifically, we examine which method-specific beliefs influence subsequent adoption of particular hormonal methods in urban and rural Kenya. We also consider the influence of past experience with specific methods.

### Study setting

The study that generated data for the current analysis was conducted in two sites; 1) one urban site comprising of two Nairobi slums (Viwandani and Korogocho) which are part of the Nairobi Urban Health and Demographic Surveillance System (NUHDSS), managed by the African Population and Health Research Center (APHRC); and 2) one rural site of Homa Bay County in Western Kenya. Although the two study populations are in the same country, they differ radically in ethnicity, education, and occupation. Details of the study populations have been published [21] but, in brief, the slum populations in Nairobi are ethnically diverse, highly mobile and have poor health outcomes partly due to pervasive poverty and unmet service-provision needs, including poor housing, poor sanitation facilities, and poor health services.

Contraceptive use among married women is relatively high (53%) and total fertility is 3.5 births per woman [22]. The population of Homa Bay is predominantly composed of the Luo ethnic group and the main occupation is subsistence agriculture and fishing. The county has a high fertility rate of 5.2 and a high unmet need for family planning (26%), with less than half (47%) of married women using contraception [23]. It also has a much higher level of HIV infection than Nairobi (20% and 4%, respectively) [24]. Understanding contraceptive perceptions and how they bear on method choice can strengthen the provision of family planning counseling.

## Methods

### Data

Data are from a two-year longitudinal study, *Improving Measurement of Unintended Pregnancy and Unmet Need for Family Planning*, conducted among married or cohabiting women aged 15–39 years at the time of recruitment [21]. In Nairobi, respondents were selected randomly from the NUHDSS database. In Homa Bay, twelve sub-locations were selected at random from three sub-counties, namely Ndhiwa, Rachuonyo North and Rachuonyo South. Household listings were then done, and eligible women were randomly selected from the lists. In each site, the study targeted a sample of 2,600 women to detect a 30% difference in reproductive outcomes (pregnancy, use and non-use of contraceptives) between study rounds at 95% confidence level and 80% power, and accounting for 45% attrition rate. Only married or cohabiting women aged 15–39 years at the time of recruitment were eligible to participate. The restriction on the upper age limit was deliberate to allow follow-up of women when they were more likely to be at risk of pregnancy compared to unmarried, non-cohabiting or older women.

Structured interviews were conducted by trained female interviews in local languages with eligible women at baseline survey (round 1), and 12-month follow-up (round 2). In Homa Bay, but not in Nairobi, a third round was conducted but results from this final round are not considered in this paper. Respondents who were sterilized or who self-reported as infecund, as well as those who had separated or been widowed between listing and interview, were excluded from follow-up owing to their reduced risk of pregnancy. In Nairobi, a total of 2812 women were interviewed at baseline, while 2195 women completed the second-round interview, representing 78 percent of those who were interviewed at baseline. In Homa Bay, a total of 2424 women were interviewed during the first round, while 2083 women completed the second round representing 86 percent of those interviewed in round 1.

The survey collected information on women's background, reproduction, fertility preferences, and current and past use of methods. One section of the questionnaire collected information on women's perceptions concerning method attributes. Questions were asked about eight methods but here we present data only for injectables, pills and implants, a restriction dictated by the fact that insufficient numbers adopted other methods to sustain analysis in both sites. All women who had heard of the specific method were asked about their perceptions of eleven method attributes, regardless of whether or not they had ever used the method. Women were first asked whether the method was easy to obtain and whether it was, in their opinion, easy to use. Perceived effectiveness was ascertained by asking respondents whether or not they considered the method to be "very effective at preventing pregnancy". Five items related to health concerns and safety. Specifically, women were asked whether they thought the method was likely to cause: (a) serious health problems; (b) unpleasant side effects; (c) disruption to regular menses; (d) long term infertility; (e) dangers if used for a long time without taking a break. Note that beliefs b and c are valid while beliefs a, d, and e are erroneous. Because of evidence on the importance of social influences [25], women were asked how many members, if any, of their social network had used the method and whether their experience

had been satisfactory. In the regression analysis, responses to these two questions were combined to form a binary variable: some members of the social network had tried the method and were satisfied versus other possibilities. Women's perception of their partners' approval of the method was also ascertained; in the analysis, don't know responses were combined with disapproval. The final variable concerned respondents' past use of the method and the degree of satisfaction. Respondents were classified into three categories; used and satisfied; used and dissatisfied or mixed opinion; and never used. The precise wording and sequence of questions can be found at (http://stepup.popcouncil.org/library/STEPUP_questionnaire_31072016.pdf).

## Analytical approach

Analysis on method choice is based on a subset of women who were not using any method at baseline but who had adopted implants, injectables or pills as of 12-month follow-up or round 2 (n = 221 in Nairobi and n = 197 in Homa Bay). Fig 1, a flow diagram, shows how the analytical sample was derived. Non-users at baseline were a mixture of pregnant women, women who wanted a child soon and those with unmet need for family planning. We use descriptive statistics to compare the baseline characteristics of women who: 1) adopted implants, injectables or pills; 2) adopted other methods (including female sterilization, IUD, Condom, Lactational Amenorrhea method, Rhythm and withdrawal methods; and 3) non-adopters at 12-month follow-up. We also examined baseline method-specific beliefs about each of the three methods—injectables, pills or implants—adopted as well as past use and satisfaction.

For the key statistical analysis of factors that influence method choice, we have two types of variables: 1) characteristics of the woman, including age, level of education, fertility preferences and baseline pregnancy status, which vary only between respondents; and 2) the set of method-specific beliefs, which vary between respondents and between methods. As these two types of variables are difficult to accommodate in conventional multinomial models, we used McFadden's conditional discrete choice model to determine the influence of each method-specific attribute on the likelihood of using injectables, pills, or implants [26]. This model has been widely used in economic investigations of choice and also applied to contraceptive choices [27, 28]. This model accommodates inclusion in one regression equation of both types of variables. For each method attribute or belief, a single coefficient is obtained which represents its association with method choice. For respondents' characteristics, there are arrays of coefficients for effects on two pairwise method choices: implant versus injectable, and pill versus injectable (i.e. injectable serves as reference category).

We tested for correlation between attributes using Cramer's V test. The perception of a method having unpleasant side effects was highly correlated with the belief that it interferes with menses or causes health problems and was therefore excluded from the multivariate model. We present odds ratios and 95% confidence intervals. Because of the small numbers of women who adopted one of the three methods, confidence intervals are wide and we flag associations of borderline statistical significance ($p < .10$) in addition to the more customary p-values; all analyses were conducted using Stata$^®$ version 15.1, using the *asclogit* procedure to estimate the discrete choice regressions.

## Ethical considerations

Written informed consent was obtained from all participants during each round of the survey. Ethical approvals for the study were granted by the Observational/Interventions Research Ethics Committee of London School of Hygiene and Tropical Medicine (11331), the Institutional Review Board of the Population Council (Protocol-644), as well as the African Medical and Research Foundation (AMREF) Ethics and Scientific Review Committee for the Nairobi site

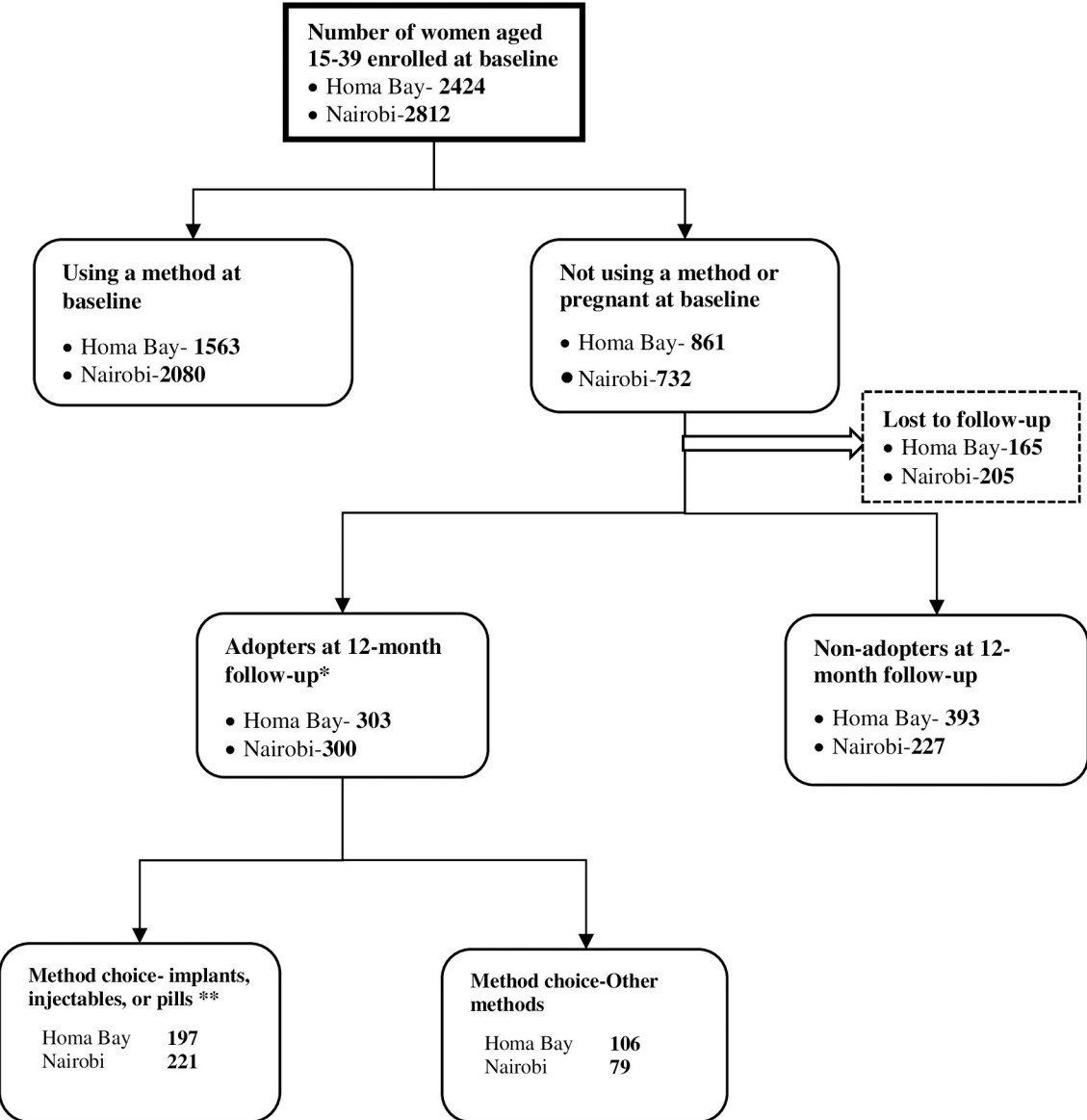

**Fig 1. Flow diagram indicating method choice at 12-month follow-up.** *Adopters of any method among women who were not using any contraceptive or pregnant but were aware of pills, injectables, and implant at baseline. ** Adopters of pills, injectable or implants among women were not using any method or pregnant but were aware of these three methods at baseline.

(P246/2016), and Kenyatta National Hospital/University of Nairobi Ethics and Research Committee for Homa Bay site (P564/07/2016). The National Commission for Science, Technology and Innovation granted the research permit for conducting the study in Kenya (NACOSTI/P/16/8900/1485).

## Results

### Background characteristics

In both sites, the majority of women who were not using any method at baseline but adopted implants, injectables or pills at 12-month follow up were aged 25–39 (75% in Nairobi; 61% in

Homa Bay) and wanted no more child(ren) or to delay for 5 or more years before having another child (59% in Nairobi; 53% in Homa Bay). The proportion of women having secondary or above level of education who adopted implants, injectables or pills at 12-month follow-up was higher in Nairobi (43%) than Homa Bay (28%). In Nairobi, there were significant associations between method choice and age, level of education and fertility preference. For example, the proportion of older women (aged 25–39) was significantly higher among non-adopter (87%) than among those who adopted implants, injectables or pills (75%), or those who adopted other methods (77%). By contrast, in Homa Bay, no significant associations between method choice and age or level of education were found. However, method choice and fertility preference were related: the proportion of women wanting a child soon or within two years was higher among non-adopters (34%) than among those who adopted implants, injectables or pills (12%), or those who adopted other methods (17%) (Table 1).

## Contraceptive method choice at follow-up

Fig 2 presents the method of contraception adopted at 12-month follow-up among women who were not using a method or pregnant at the time of baseline survey and were aware of pills, injectables and implants. In Nairobi, about 25% (N = 130) of women adopted injectables compared to 18% in Homa Bay (N = 124). In both sites, implants were chosen by about 9% of women. Conversely, pills were a more common choice in Nairobi (8%, N = 43) than in Homa Bay (2%, N = 14). It may also be noted that condoms were adopted more frequently in Homa Bay than in Nairobi, reflecting the high level of HIV infection in the rural site. Of non-users at baseline, 43% and 56% of the women in Nairobi and Homa Bay, respectively, remained non-users at follow-up or were pregnant.

**Table 1. Characteristics of married or cohabiting women aged 15–39 years by use status and method choice at follow up among non-users at baseline.**

| | Nairobi | | | | Homa Bay | | | |
|---|---|---|---|---|---|---|---|---|
| | Adopted implants, injectables or pills | Adopted other methods^ | Non-adopters | p-value | Adopted implants, injectables or pills | Adopted other methods^ | Non-adopters | p-value |
| **Age** | | | | | | | | |
| 15–24 | 24.9 | 22.8 | 12.3 | | 38.6 | 42.5 | 34.9 | |
| 25–39 | 75.1 | 77.2 | 87.7 | 0.002 | 61.4 | 57.6 | 65.1 | 0.312 |
| **Highest level of education** | | | | | | | | |
| No education/some primary | 25.8 | 20.3 | 30.0 | | 44.2 | 42.5 | 47.8 | |
| Completed primary | 31.2 | 32.9 | 37.9 | 0.073 | 27.9 | 31.1 | 32.3 | 0.213 |
| Secondary+ | 43.0 | 46.8 | 32.2 | | 27.9 | 26.4 | 19.9 | |
| **Fertility preference** | | | | | | | | |
| Want soon/want within 2 years/undecided | 29.0 | 31.7 | 59.9 | | 12.2 | 17.0 | 34.1 | |
| Want to wait 2–4 years | 12.2 | 22.8 | 17.2 | <0.001 | 24.4 | 28.3 | 19.6 | <0.001 |
| Want to wait 5+ years | 31.7 | 16.5 | 9.7 | | 26.9 | 21.7 | 15.0 | |
| Wants no more | 27.2 | 29.1 | 13.2 | | 36.6 | 33.0 | 31.3 | |
| **Pregnant at baseline** | | | | | | | | |
| Not pregnant | 92.8 | 95.5 | 93.7 | | 93.0 | 85.6 | 87.6 | |
| Pregnant | 7.2 | 4.5 | 6.3 | 0.196 | 7.0 | 14.4 | 12.4 | <0.001 |
| **Number of women (N)[b]** | **221** | **79** | **227** | | **197** | **106** | **393** | |

Note: ^ Includes female sterilization, IUD, condom, lactational Amenorrhea method, rhythm and withdrawal methods;

[b]Ns includes only women who were not using a method or pregnant and were aware of pills, injectables and implants at the time of baseline survey

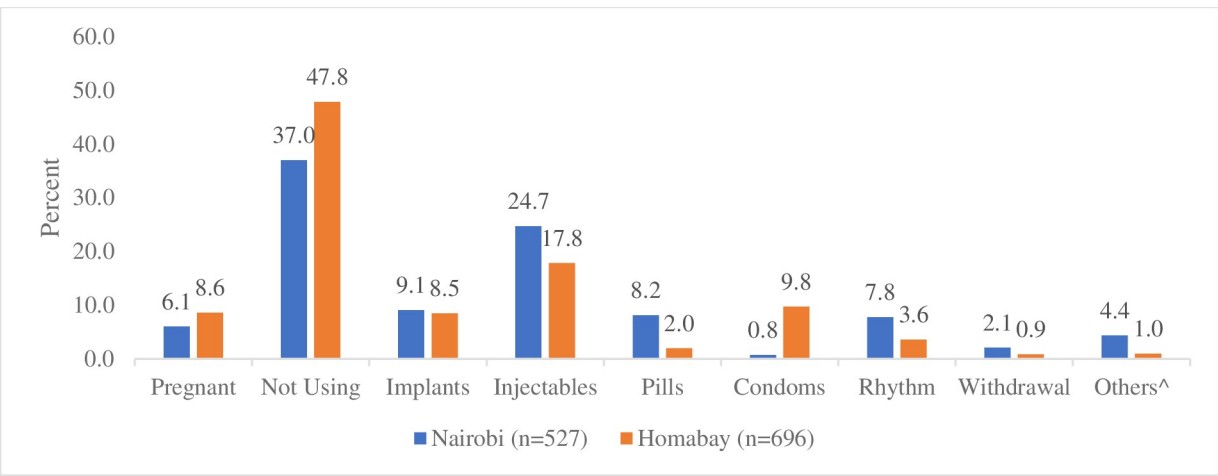

**Fig 2. Method choice at 12-month follow-up by site**\*. \* Among women not using any method at baseline but were aware of pills, injectables and implant. ^Others include female sterilization, IUD, Condom, Lactational Amenorrhea method, Rhythm and withdrawal methods.

## Beliefs about contraceptive attributes

Table 2 summarizes views on, and experience with pills, injectables, and implants among women who were not using any method at the time of baseline survey but adopted one of these methods at 12-month follow-up. In both sites, a large majority of women (range 80–98%) perceived that all three methods were easy to obtain. In terms of effectiveness and ease of use, pills were ranked lower than injectables and implants.

There were substantial differences among methods in perceived health-related concerns, and these patterns varied by site. In Nairobi, lack of concerns about unspecified serious health problems, interference with menstruation, unpleasant side effects, and infertility were more common for pills than for injectables and implants. In contrast, the proportion of women who believed that the method was safe for long term use was lowest for pills and highest for implants. In Homa Bay, positive evaluations of pills, injectables and implants on health-related effects were much lower than in Nairobi, with modest variation among the three methods. For example, a minority of women (16–28%) perceived that pills, injectables or implants do not interfere with menstruation or are safe for long-term use without taking a break. However, beliefs about fertility impairment were similarly positive in Homa Bay and Nairobi.

In both sites, the proportion of women who perceived that their husbands approved of the method was higher for injectables than for pills and implants. In addition, the proportion of women reporting knowing someone in their social network who had used the method was significantly higher for injectables and implants (range 88–96%) than for pills. Perceived satisfaction among social network members was lower for the pill than for injectables and implants. While large majorities of women in both sites had never used pills or implants (range 62%-78%), most (77% in Nairobi and 65% in Homa Bay) had previously used injectables. Among past users of the three methods, typically about half expressed satisfaction, though this proportion was higher for injectables than the other methods.

## Conditional logit analysis of method choice

Tables 3 and 4 present results from the conditional logit regression analysis of contraceptive method choice at 12-month follow-up among non-users at baseline. The upper two panels show the effects of method beliefs and satisfied past use on choice of injectables, implants or

**Table 2. Among women who were not using any method at baseline and adopted pills, injectables or implants at 12-month follow-up, the percentage with specific perception about these methods.**

| | Nairobi | | | | Homa Bay | | | |
|---|---|---|---|---|---|---|---|---|
| | Pills | Injectables | Implants | p-value | Pills | Injectables | Implants | p-value |
| **Method attribute** | % | % | % | | % | % | % | |
| **Convenience/effectiveness** | | | | | | | | |
| Easy to obtain | 93.2 | 98.2 | 87.3 | <0.001 | 79.9 | 92.9 | 85.3 | <0.001 |
| Effective at preventing pregnancy | 67.0 | 88.2 | 90.1 | <0.001 | 59.8 | 92.0 | 87.1 | <0.001 |
| Easy to use | 40.3 | 89.1 | 77.8 | <0.001 | 30.8 | 83.9 | 73.2 | <0.001 |
| **Health effects concerns** | | | | | | | | |
| Does not causes serious health problems | 56.1 | 45.7 | 37.1 | <0.001 | 22.8 | 29.5 | 30.8 | 0.126 |
| Does not interfere with menstruation | 55.2 | 20.4 | 36.7 | <0.001 | 21.4 | 17.9 | 19.6 | 0.636 |
| Does not causes unpleasant side effects | 47.1 | 38.9 | 36.2 | 0.054 | 21.0 | 32.6 | 28.1 | 0.021 |
| Safe for long-term use (without a break) | 26.2 | 32.1 | 44.8 | <0.001 | 16.1 | 23.7 | 27.7 | 0.011 |
| Does not cause infertility | 84.6 | 76.0 | 72.9 | 0.009 | 72.8 | 73.2 | 76.3 | 0.643 |
| **Social** | | | | | | | | |
| Perceived husband approval of method | 62.4 | 80.1 | 62.4 | <0.001 | 46.0 | 65.2 | 51.3 | <0.001 |
| Have a friend/relative/neighbor who have used the method | 82.4 | 96.4 | 94.6 | <0.001 | 68.3 | 89.3 | 87.5 | <0.001 |
| Friends/relatives/neighbors are satisfied with method† | 43.0 | 62.0 | 55.7 | <0.001 | 32.8 | 42.8 | 55.6 | 0.022 |
| **Past use and satisfaction** | | | | | | | | |
| Never used | 62.4 | 23.1 | 75.1 | | 75.9 | 34.8 | 77.7 | |
| Used and satisfied | 19.9 | 47.1 | 13.1 | <0.001 | 10.3 | 35.3 | 12.1 | <0.001 |
| Used and dissatisfied | 17.7 | 29.9 | 11.8 | | 13.8 | 29.9 | 10.3 | |
| **Number of Women (N)^** | **221** | **221** | **221** | | **197** | **197** | **197** | |

Note: *p < .05. **p < .01. ***p < .001;

†Among those who reported knowing someone who had used the method; ^ Ns are total number of women who were not using any method but were aware of pills, injectables and implant at baseline

pills, while the lower panels show the effects of respondent characteristics on choosing the method (implant or pill) versus the base alternative method (injectable). The left-hand column shows the crude, or unadjusted, associations with method choice, and the right-hand column shows the adjusted associations.

Considering first the results for Nairobi, satisfied past use, perceived husband approval, ease of use, and absence of long-term fertility impairment had the largest unadjusted associations with method choice, with odds ratios of 2.6 or more. Effectiveness and absence of serious health problems were more weakly, but significantly (p < .05), associated with method choice, with odds ratios of 1.97 and 1.68, respectively. The association of the social network variable and method choice was weaker (OR = 1.4; p < .10). The results from the adjusted model, after the omission of serious side effects for reasons outlined earlier, were not radically different from the unadjusted results. Satisfied past use, perceived husband approval of the method, ease of use and no long-term fertility impairment retained strong associations with method choice. The only other association, of borderline statistical significance, was a small negative effect of perceived safety to use for a long time (AOR = 0.61; p < .10).

The unadjusted results for Homa Bay showed some similarities with those for Nairobi but also differences. As in Nairobi, satisfied past use and ease of use had the strongest associations with method choice, but, unlike Nairobi, absence of serious side effects, had significant positive association, with an odds ratios of 1.8. In addition, perceived husband approval of the method and long-term fertility impairment were more weakly associated with method choice than in Nairobi.

**Table 3. Odds ratios (and 95% confidence intervals) from conditional logit regression analyses assessing women's likelihood of using injectable, pill or implants at 12-month follow-up by perceived method attributes, past use and satisfaction, and selected characteristics -Nairobi.**

| | Crude OR [95%CI] | AOR [95%CI] |
|---|---|---|
| **Effects of Method Attributes** | | |
| Easy to obtain | 1.29 [0.54–3.10] | 0.70 [0.25–1.96] |
| Effectively prevents pregnancy | 1.97 [1.10–3.55] * | 1.00 [0.48–2.12] |
| Easy to use | 3.71 [2.11–6.52] *** | 3.00 [1.52–5.94] ** |
| Absence of serious health problems | 1.68 [1.08–2.59] * | 1.03 [0.60–1.76] |
| No interference with menstruation | 1.27 [0.83–1.93] | 1.02[0.60–1.72] |
| Absence of unpleasant side effects | 1.23 [0.82–1.86] | - |
| Safe for long time use (without a break) | 1.16 [0.72–1.88] | 0.60 [0.33–1.08] ± |
| No long-term fertility impairment | 2.62 [1.43–4.82] ** | 2.60 [1.26–5.39] ** |
| Social network tried and satisfied | 1.42 [0.96–2.08] ± | 0.82 [0.50–1.36] |
| Perceived husband approval of the method | 3.42 [1.89–6.18] *** | 2.08 [1.05–4.11] * |
| **Past use and satisfaction** (Ref: Never used) | | |
| Past user and satisfied | 4.49 [2.84–7.12] *** | 3.68 [2.16–6.28] *** |
| Past user and dissatisfied/mixed/neither | 1.47 [0.83–2.60] | 1.36 [0.72–2.56] |
| **Effects on Choice of Pill (vs. Injectable)** | | |
| **Age group** (Ref: 15–24 years) | | |
| 25–39 years | 1.43 [0.60–3.39] | 1.00 [0.35–2.84] |
| **Educational attainment** (Ref: no education/ incomplete primary) | | |
| Completed primary | 2.21 [0.77–6.37] | 1.67 [0.49–5.72] |
| Secondary+ | 2.63 [0.97–7.08] ± | 2.16 [0.69–6.81] |
| **Fertility Preference** (Ref: Want to soon/want within 2 years/undecided) | | |
| Want to wait 2–4 years | 0.22 [0.05–1.07] ± | 0.35 [0.06–2.11] |
| Want to wait 5+ years | 0.33 [0.13–0.84] * | 0.46 [0.14–1.54] ± |
| Want no more | 1.10 [0.47–2.56] | 1.75 [0.60–5.08] |
| **Pregnant at baseline** (Ref: No) | 0.46 [0.22–0.96] * | 0.62 [0.24–1.61] |
| **Effects on Choice of Implant (vs. Injectable)** | | |
| **Age group** (Ref: 15–24 years) | | |
| 25–39 years | 0.72 [0.35–1.49] | 0.55 [0.23–1.30] |
| **Educational attainment** (Ref: no education/ incomplete primary) | | |
| Completed primary | 1.08 [0.46–2.53] | 0.82 [0.31–2.18] |
| Secondary+ | 0.88 [0.39–1.99] | 0.62 [0.24–1.58] |
| **Fertility Preference** (Ref: Want to soon/want within 2 years/undecided) | | |
| Want to wait 2–4 years | 1.02 [0.34–3.00] | 1.01 [0.26–3.84] |
| Want to wait 5+ years | 0.69 [0.29–1.68] | 0.74 [0.26–2.07] |
| Want no more | 1.35 [0.55–3.30] | 1.49 [0.50–4.29] |
| **Pregnant at baseline** (Ref: No) | 0.83 [0.43–1.61] | 0.66 [0.31–1.56] |
| **Number of Observations** | **663** | **663** |
| **Number of cases (N)** | **221** | **221** |

Note: *p < .05.

**p < .01.

***p < .001 ±p<0.10; OR Odds Ratio; AOR Adjusted Odds Ratio; Ref-Reference category

The adjusted results for Homa Bay showed fewer statistically significant associations than in Nairobi. Satisfied past use remained the strongest predictor while the association with ease of use was large, with an odds ratio of 2.0 but with a lower level of statistical certainty (p < .10).

**Table 4. Odds ratios (and 95% confidence intervals) from conditional logit regression analyses assessing women's likelihood of using injectable, pill or implants at 12-month follow-up by perceived method attributes, past use and satisfaction, and selected characteristics-Homa Bay.**

|  | Crude OR[95%CI] | AOR[95%CI] |
|---|---|---|
| **Effects of Method Attributes** |  |  |
| Easy to obtain | 1.71[0.84–3.46] | 1.02[0.45–2.31] |
| Effectively prevents pregnancy | 1.38[0.67–2.80] | 0.82[0.34–1.96] |
| Easy to use | 2.27[1.33–3.86] ** | 1.96[0.96–3.98] ± |
| Absence of serious health problems | 1.34[0.81–2.19] | 1.02[0.51–2.03] |
| No interference with menstruation | 1.53[0.85–2.77] | 0.83[0.39–1.80] |
| Absence of unpleasant side effects | 1.79[1.09–2.90] * | - |
| Safe for long time use (without a break) | 1.32[0.69–2.52] | 0.94[0.47–1.91] |
| No long-term fertility impairment | 1.70[0.86–3.38] | 1.69[0.70–4.05] |
| Social network tried and satisfied | 1.18[0.74–1.87] | 0.78[0.44–1.37] |
| Perceived husband approval of the method | 1.71[0.93–3.13] ± | 1.00[0.48–2.10] |
| **Past use and satisfaction** (Ref: Never used) |  |  |
| Past user and satisfied | 2.83[1.64–4.80] ** | 2.56[1.31–4.99] ** |
| Past user and dissatisfied/mixed/neither | 1.29[0.75–2.20] | 1.26[0.67–2.36] |
| **Effects on Choice of Pill (vs. Injectable)** |  |  |
| **Age group** (Ref: 15–24 years) |  |  |
| 25–39 years | 3.66[0.78–17.00] ± | 3.45[0.67–17.83] |
| **Educational attainment** (Ref: no education/ incomplete primary) |  |  |
| Completed primary | 1.19[0.30–4.70] | 1.03[0.23–4.67] |
| Secondary+ | 1.72[0.46–6.30] | 1.27[0.28–5.67] |
| **Fertility Preference** (Ref: Want to soon/want within 2 years/undecided) |  |  |
| Want to wait 2–4 years | 0.31[0.05–2.07] | 0.26[0.03–1.98] |
| Want to wait 5+ years | 0.95[0.20–4.52] | 0.95[0.17–5.22] |
| Want no more | 0.46[0.09–2.30] | 0.38[0.07–2.18] |
| **Pregnant at baseline** (Ref: No) | 0.80[0.24–2.74] | 0.92[0.24–3.52] |
| **Effects on Choice of Implant (vs. Injectable)** |  |  |
| **Age group** (Ref: 15–24 years) |  |  |
| 25–39 years | 0.72[0.39–1.00] | 0.80[0.35–1.80] |
| **Educational attainment** (Ref: no education/ incomplete primary) |  |  |
| Completed primary | 0.77[0.36–1.00] | 0.76[0.30–1.87] |
| Secondary+ | 1.15[0.55–2.39] | 1.22[0.51–2.93] |
| **Fertility Preference** (Ref: Want to soon/want within 2 years/undecided) |  |  |
| Want to wait 2–4 years | 1.13[0.34–3.75] | 0.90[0.21–3.77] |
| Want to wait 5+ years | 2.29[0.72–7.27] | 2.03[0.51–8.08] |
| Want no more | 1.53[0.50–4.70] | 2.16[0.57–8.61] |
| **Pregnant at baseline** (Ref: No) | 1.49[0.79–2.82] | 1.22[0.57–2.61] |
| **Number of Observations** | **591** | **591** |
| **Number of cases (N)** | **197** | **197** |

Note: *p < .05.

**p < .01.

***p < .001 ±p<0.10; OR Odds Ratio; AOR Adjusted Odds Ratio

We assessed the alternative specifications of the model. Specifically, we re-ran the analysis for Nairobi without partner's approval because of the possibility that this variable might act as a proxy for the respondent's overall view of the method. We also re-analysed data for both

sites without satisfied past use because satisfaction itself is in part a function of perceptions of methods [29]. These alternative specifications made only small differences in the pattern of results in Tables 3 and 4 and are not shown.

The lower panels of Tables 3 and 4 show the effects of respondent characteristics on method choice, first the choice of pill versus injectable and then the choice of implant versus injectable. Confidence intervals were wide and few associations can be stated with statistical confidence. The results suggest that better-educated women were more likely to choose pills over injectables in both sites than their counterparts, as were older women in Homa Bay. With regard to the choice of implants over injectables, the only result of note was the greater likelihood of choosing implant among women wanting no more children compared to those who wanted a child although the result was not significant in Nairobi. We checked the possibility that women with a recent birth and thus likely to be breastfeeding and amenorrheic at follow-up might make different method choices from other women by comparing those who were pregnant at baseline with non-pregnant women but found no significant difference in the multivariate model.

## Discussion

In these two study populations in Kenya, beliefs about pills, injectables and implants among non-users were generally negative. With the partial exception of the pill in Nairobi, the majority thought that each method was likely to cause serious health problems, unpleasant side effects, menstrual disruption, and would be unsafe for long-term use. Nevertheless, within 12 months more than a third (42%) of the women in Nairobi and a quarter (28%) in Homa Bay were using one of these three methods. It thus appears that, for many women, the desire to prevent pregnancy overrides concerns about methods.

The central topic addressed in this paper is the effect of beliefs about a method, and past experience with a method, on its subsequent use. While it seems obvious that beliefs will be related to method choice, very little is known about the relative importance of specific beliefs in predisposing women to choose one method over another. The results reveal some expected links but also some surprises. As expected, satisfactory past experience with a method was the strongest predictor of method choice in both sites. Contraceptive use has been relatively high in Nairobi and Homa Bay for some years and the majority of women who were non-users at baseline had prior experience of injectables and substantial minorities had used pills and implants. Among past users, about half reported satisfaction with the method, though the ratio of satisfied to dissatisfied users was higher for injectables in the Nairobi sample [29]. A study based on analysis of Homa Bay round 3 data shed light on the causes of dissatisfaction with injectables and implants [30]. The majority of past users of these methods reported side effects that affected menses and about 40% reported non-bleeding side effects such as dizziness and stomach pain or cramps [30]. Dissatisfaction was low among women who reported no side effects and was highest among those who experienced both menstrual and other side effects [30]. Among various types of bleeding side effects, heavy or prolonged bleeding was regarded as the most serious and dissatisfaction was highest among women with this experience [30].

This evidence of the strong influence of past experiences with specific methods and subsequent method choice is consistent with findings that much unmet need for family planning arises from women who have used a modern method in the past [31]. The range of modern methods commonly used by married women in Kenya, and elsewhere in Africa, is narrow, being dominated by injectables with an increasing contribution from implants. We speculate that women who have unsatisfactory experiences with both methods are particularly likely to abandon contraception and experience unintended pregnancies. Appropriate programmatic

responses have been much discussed: widen method choice and improve counselling that fore-warns women of likely side effects and alerts them to the availability of treatment or method-switching [32].

The second strong and statistically significant influence on method choice in both sites was the belief that the method is easy to use. This link acts to disfavour choice of pills. Whereas large majorities of women thought that injectables and implants were easy to use, this propor-tion dropped to 40% for pills in Nairobi and was even lower in Homa Bay. That many women in Kenya consider the pill difficult to use goes some way to explaining the rise in injectable use, from 7% in 1993 to 29% in 2015, whereas the use of pills has stagnated between 7% and 10% over this period. It is also of interest that the results contain tentative evidence that pills were favoured by well-educated women, who may find adherence to the discipline of the daily pill regime easier than their less-educated counterparts. The education-pill relationship may also explain the greater uptake of this method among the better educated Nairobi sample than in Homa Bay. However, the result for Homa Bay may not be conclusive due to the small sample size.

Some marked differences in results between the two sites are apparent. The belief that a method would not cause long-term fertility impairment and perceived approval of the partner were strong predictors of method choice in Nairobi but not in Homa Bay. Fear of fertility impairment is likely to be less of a concern for women who want no more children than among those wanting additional children. Among non-users at baseline, 37% in Homa Bay wanted to limit family size compared with 27% in Nairobi. This difference may well be part of the explanation for the contrasting result between the two sites.

The inter-site contrast in the association of perceived husband approval and method choice is puzzling. It implies that husbands are less involved in contraceptive decisions in the rural than the urban site. Women in Homa Bay were more likely to give 'don't know' responses than those in Nairobi (about 10% versus 1%) when asked about perceived husband approval of the method. Although 'don't know' responses were reclassified as disapproval, this is unlikely to account for the contrast—most past users who gave 'don't know' responses were also dissat-isfied with the method. Qualitative research suggested that clandestine use of contraception is common in Homa Bay because of hostility or indifference of the partner but no similar evi-dence is available for Nairobi [23]. It is also the case that polygyny is much more common in Homa Bay (25%) than in Nairobi (5%) and the existence of co-wives may create social distance between partners and reduce discussion of matters such as contraception [33]. Thus, possible but speculative reasons exist for the inter-site difference.

The lack of importance of the beliefs about menstrual disturbance and safety for long-term use on method choice in both sites should also be noted. Contrary to our findings, the research literature shows that contraceptive-induced menstrual irregularities have a major influence on method satisfaction and discontinuation [34]. Reactions to contraception-related changes in bleeding also vary widely between populations. Women may value menstrual regularities, that is, seeing menstrual disturbances as a reflection of poor health, potential challenge in their rela-tionships, and potential indicator of later infertility [34]. One possible explanation for the sur-prising absence of associations between beliefs about menstrual disturbance and method choice in our study is that the two dominant methods, injectables and implants, have rather similar effects on bleeding.

Negative results are as important as positive ones. Large majorities of women in both sites considered all three methods "easy to obtain" and this attribute was unrelated to method choice. Before adjustment, the belief that a method was effective was related to method choice in Nairobi but not in Homa Bay, however, after adjustment, the association attenuated. While beliefs about the effectiveness of injectables and implants were very high, they were much

lower for pills but this factor appears to be a less important consideration for method choice than ease of use, which was also much lower for pills than the other methods. Similarly, the association between method choice and beliefs about serious unspecified health problems disappeared after adjustment for other factors, including specific health-related beliefs such as fertility impairment and safety for long-term use. In view of extensive evidence of the influence of social networks on contraceptive decisions [12, 35], the biggest surprise was the lack of association with the contraceptive experience of others in the woman's social network. Nevertheless, the opinions and experiences of friends and neighbours may matter as a key source of the beliefs about methods that show net associations, particularly for women who have never tried a particular method [12, 35].

The main strength of this study is its longitudinal nature which enhances explanatory plausibility: baseline data on beliefs were collected and used as predictors of method-specific uptake after an interval of 12 months among women who were using no method or pregnant at baseline.

It also has limitations. We included only three methods, pills, injectables and implants, though they do comprise a large share of the modern method mix. For instance, it would have been particularly interesting to assess beliefs that incline women to adopt traditional methods, but sample sizes precluded this option. In the time interval between baseline and 12-month follow up, it is possible that beliefs changed, perhaps because of counselling by a health provider. We also missed in the analysis episodes of use that started but ended between baseline and follow up, though it is very unlikely that the small number of such episodes would have affected the results.

## Conclusions

In these two study populations with relatively high levels of personal past experience of hormonal methods of contraception, satisfied past use of a method was a major determinant of future method choice. The other major influence, common to both sites, was the perception that a method is easy to use., The erroneous belief about long-term fertility impairment and perceived husband approval had a strong influence on method choice among women living in Nairobi slums but not in the rural Western Kenyan site. Thus, the relative importance of beliefs in predisposing women to choose one method over another appears to be conditioned by the social context. There is need for family planning counseling programmes to respond to erroneous beliefs and misconceptions about contraceptives.

## Acknowledgments

We acknowledge the data collection teams in both study sites. We are also thankful to the study participants who freely provided information and community leaders in the respective study sites.

## Author Contributions

**Conceptualization:** George Odwe, Francis Obare, Kazuyo Machiyama, John Cleland.

**Data curation:** Yohannes Dibaba Wado, Francis Obare.

**Formal analysis:** George Odwe.

**Investigation:** Francis Obare, Kazuyo Machiyama.

**Methodology:** George Odwe, Kazuyo Machiyama, John Cleland.

**Supervision:** George Odwe, Kazuyo Machiyama, John Cleland.

**Validation:** George Odwe, Francis Obare, Kazuyo Machiyama.

**Writing – original draft:** George Odwe, John Cleland.

**Writing – review & editing:** George Odwe, Yohannes Dibaba Wado, Francis Obare, Kazuyo Machiyama, John Cleland.

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
