## [Decision Letter · Decision Letter 0]

5 Feb 2021

PONE-D-20-34145

Method-specific beliefs and subsequent contraceptive method choice: results from a longitudinal study in urban and rural Kenya

PLOS ONE

Dear Dr. Odwe,

Thank you for submitting your manuscript to PLOS ONE. After careful consideration, we feel that it has merit but does not fully meet PLOS ONE’s publication criteria as it currently stands. Therefore, we invite you to submit a revised version of the manuscript that addresses the all points raised by both reviewers during the review process.

We look forward to receiving your revised manuscript.

Kind regards,

Philip Anglewicz, PhD

Academic Editor

PLOS ONE

Journal Requirements:

2)  We note that you have indicated that data from this study are available upon request. PLOS only allows data to be available upon request if there are legal or ethical restrictions on sharing data publicly. For more information on unacceptable data access restrictions, please see http://journals.plos.org/plosone/s/data-availability#loc-unacceptable-data-access-restrictions.

Reviewers' comments:

Reviewer's Responses to Questions

**Comments to the Author**

1. Is the manuscript technically sound, and do the data support the conclusions?

Reviewer #1: Yes

Reviewer #2: Yes

2. Has the statistical analysis been performed appropriately and rigorously? 

Reviewer #1: Yes

Reviewer #2: Yes

3. Have the authors made all data underlying the findings in their manuscript fully available?

Reviewer #1: Yes

Reviewer #2: Yes

4. Is the manuscript presented in an intelligible fashion and written in standard English?

Reviewer #1: Yes

Reviewer #2: Yes

5. Review Comments to the Author

Reviewer #1: PLoS One Review

Title: Method-specific beliefs and subsequent contraceptive method choice: results from a longitudinal study in urban and rural Kenya

Manuscript Number: PONE-D-20-34145

OVERALL FEEDBACK: This study examines an important and timely topic—the relationship between contraceptive beliefs and adoption of specific contraceptive methods. The authors explore a range of different characteristics specific to the three methods explored as outcomes (implants, injectables and pills), which builds substantially on a currently limited literature base. Clarification of the distinct study samples, derived from the cohort study, would improve the methods section; use of a study selection flowchart is recommended. As described in the feedback for the methods section, it is unclear why the authors used follow-up data from different survey rounds when comparing Nairobi and Homa Bay, thereby inducing systematically different follow-up times between the samples. Given the authors’ inclusion of women who were pregnant at enrollment (more than one-quarter of non-users at baseline in both samples), I expected to see some assessment of or adjustment for postpartum status, as this is likely to shape how women act on beliefs to make decisions about contraception. Other specific suggestions, which I hope will improve the clarity and contribution of this important work, are itemized by section. A few grammatical errors, verb tenses, and missing words are also present throughout the manuscript and require attention before finalization.

ABSTRACT

• On line 29, the authors use “round 2” to talk about follow-up data collected as part of the mentioned longitudinal study. This terminology is confusing because the study design has not yet been described. Please review the abstract accordingly.

INTRODUCTION

• On line 54, the authors state: “Fear of side effects, damage to health, menstrual disruption, and long-term infertility have been widely documented, including for Kenya [6-8].” Sentence should be clarified to state that these are the barriers mentioned in the previous sentence.

MATERIALS AND METHODS

• On line 94, please define acronym “NUHDSS” at first use.

• Lines 103-110 describe how the authors arrived at the analytic sample. A figure would be helpful here, particularly since a broader sample is included in Tables 1-2, compared to the final analytic sample for the specific contraceptive outcomes in Tables 3-5. This would be especially helpful given the difference in follow-up times between Nairobi and Homa Bay samples.

• On lines 103-106, the authors describe the cohort study samples from the two sites. While there was no “round 3” conducted in Nairobi, why did the authors choose to use the different follow-up rounds for the two sites? Why not just restrict them both to round 2, given that the follow-up time between the baseline and follow-up samples would be more comparable? In the methods description, it does seem like round 2 data is available in Homa Bay, so this should be used instead of round 3. If not adjusting for time since baseline belief in each model, both models should at least include comparable samples in terms of follow-up time.

• On line 112, the authors state, “…1866 women during the third round (91% of those interviewed in round 2).” This should be stated in terms of the retention from the original sample, not the round 2 sample. This recommendation can be ignored if the sample for Homa Bay is revised to include round 2 as follow-up only.

• On line 125-126, the authors state, “Note that beliefs b and c are valid while beliefs a, d,

and e are erroneous.” Why do the authors make the distinction between beliefs about contraception that are (in their mind) “valid” and others are considered “erroneous”? If these beliefs are shaped by women’s social networks, communities, etc. and, in turn, inform women’s contraceptive decision-making, then why is this distinction important? Additionally, the authors should include some mention of the large body of literature which ties contraceptive use/non-use to beliefs about contraception and infertility.

• On line 126, the authors mention, “Because of evidence on the importance of social influences…” Please cite relevant literature here or in the introduction.

• On lines 130-131, the authors mention that, “Women's perception of their partners' approval of the method was also ascertained; in the analysis, don't know responses were combined with disapproval.” Why were “don’t know” responses combined with disapproval and not approval? For what proportion of the sample did this affect? I am particularly interested in this given the significant findings discussed later in the manuscript, which show a strong influence of perceived partner disapproval and contraceptive choice in Nairobi. Was there a difference between the proportion of women who selected “don’t know” between urban and rural sites that may have related to the observed differences?

• On line 146, the word “variable” is meant to be plural (i.e., variables).

• Participants who were pregnant at baseline were included in the analysis, accounting for more than one-quarter of all non-users at baseline. Was there any assessment of if method choice was shaped by whether or not women were postpartum? I expected to see some assessment of or adjustment for postpartum status, as this is likely to shape how women act on beliefs to make decisions about contraception. I understand that sample sizes are certainly an issue, but feel strongly that postpartum status be recognized in this analysis.

RESULTS

• On line 178-179, the authors state: “In Nairobi, 74% were using a method of contraception at baseline compared with 65% in Homa Bay.” In the methods section, the analytic sample is described to be women who were not using contraception at baseline, given that the focus of the analysis is contraceptive method adoption between baseline and follow-up. It is confusing to have Tables 1-2 present data for the full sample of women from the existing cohort, as many of them are not eligible for participation in the present study since they are not “at risk” for the outcome.

o If Tables 3-5 present results for the subsample of women who were not using at baseline, but had adopted an injectable, pill or implant by follow-up, Table 1 should reflect these same populations for comparison. For example, if the authors feel it is important to retain information on the full cohort in this paper, Table 1 might show three categories in each site: contraceptive users at baseline vs. non-users at baseline who do not adopt implants, injectables or pills vs. non-users at baseline who adopt implants, injectables or pills (i.e., analytic sample for Tables 3-5, main analyses). This would make it easier, from a reader’s perspective, to compare the population of women in the analytic sample to other women surveyed.

• With the previously suggested revision, update lines 176-186 to provide comparison between the analytic sample and the other two samples derived from the same cohort.

• In Table 3, please define “ns” in the notes under the table as “non-significant”.

• Tables 4-5 should include Ns to specify analytic sample sizes.

• Clarify throughout the results—and in other sections—when “husband’s approval” is stated that this is “perceived husband approval”, as this measure is reported by the woman herself (e.g., lines 240, 260, etc.)

• Line 276 includes a typo for the word “above”

DISCUSSION

• The paragraph articulated in lines 309-317, connection between prior contraceptive use and future use relating to unmet need and programs, is excellent. Bravo!

• Starting on line 329, the authors focus the discussion on the contrast observed in the relationship between contraceptive beliefs and method-specific use across the two sites. See earlier comment in methods section about making these two samples more comparable with the use of round 2 data from Homa Bay to reflect the same follow-up times between baseline and follow-up for the two samples.

• Starting on line 336, the authors describe the perplexing association between perceived husband’s approval and method choice, comparing the two sites. See earlier comment in methods section about the proportion of women who responded “don’t know” and how they were reclassified as having partners that “disapproved” of the method. Given these inconsistent results across the two sites, I remain curious about the proportion of women in each site whose responses were reclassified and how this reclassification might have misclassified women with uncertainty about their partners’ beliefs as having

“disapproving” or hostile partners and therefore shaped the observed results.

• On lines 344-346, the authors note that, “Beliefs about menstrual disturbance and safety for long-term use were related to method choice in the rural sample but not in Nairobi”. Perhaps women in Homa Bay value menstrual regularities more than women in Nairobi, seeing menstrual disturbances as a reflection of poor health, potential challenge in their relationships, potential indicator of later infertility, etc. whereas women in Nairobi conceptualize menstrual disturbances differently (e.g., that no bleeding is good, not a sign of poor health). See Polis 2018, ‘“There might be blood”: Scoping review of women’s responses to contraceptive-induced menstrual bleeding changes’ and other articles focused on how changes in menstruation affect women differently.

• On lines 346-348, the authors highlight how the sites are distinct, indicating that their comparison might not be appropriate. Given the focus on the site-specific differences throughout the results and flagged in the abstract, this statement should come earlier in the article. Authors should clearly indicate that the two sites were selected due to the availability of data in the cohort study, etc. and not due to their comparability overall.

• I appreciate the authors’ focus on the negative results in lines 349-361. This is an important contribution—equally as valuable as the positive results, as the authors note!

• As flagged in the methods, the authors should reference literature on social networks and contraception to contextualize the results outlined in lined 358-361.

o On line 367-368, the authors state that women’s contraceptive beliefs may have changed “perhaps because of an unsatisfactory experience of a social network member”, which affected the insignificant relationships between many beliefs and later use of methods. Given the last point about the lack of observed association between social network experiences and women’s contraceptive choices, I don’t think this makes sense as a plausible explanation.

CONCLUSION

• On lines 383-384, the authors end with “Women's choices about which method to use should be based, as far as possible, on correct knowledge.” This seems like a bit of a leap. The focus should not be on women’s “correct knowledge” but instead on how valid concerns, misconceptions, fears, etc. of contraceptive attributes require attention from the field.

Reviewer #2: The paper is well written. It adds into existing literature on contraceptive method beliefs that act as facilitators or barriers to contraceptive uptake and continuation with use.

Introduction:

• The last statement on strengths should be moved to the discussion section (line 74-76)

Methods

• Lines 93/100-Provide a rationale for limiting the eligibility criteria to women aged 15-39 years? Why not women of reproductive age,15-49years?

• Line 118: How were the 11 attributes selected? The five items related to health concerns and safety have some overlap. For instance, unpleasant side effects might also include option c (disruption of regular menses). Was the tool/questions pretested before administration?

Results

• line 198-199 Was the use of condoms especially in Homa Bay mainly for contraceptives or also for dual protection?

• Were there any differences in the baseline characteristics of the women in the two counties?

• Was data collected on discontinuation or method switching between round 2 and round 3 for Homa Bay?

Discussion

• Lines 290/1-Were the women asked whether the method they chose was their preferred option?

6. PLOS authors have the option to publish the peer review history of their article (what does this mean?). If published, this will include your full peer review and any attached files.

Reviewer #1: No

Reviewer #2: No

---

## [Author Response · Author response to Decision Letter 0]

26 Mar 2021

PLoS One Review

Title: Method-specific beliefs and subsequent contraceptive method choice: results from a longitudinal study in urban and rural Kenya

Manuscript Number: PONE-D-20-34145

REVIEWER #1

OVERALL FEEDBACK: This study examines an important and timely topic—the relationship between contraceptive beliefs and adoption of specific contraceptive methods. The authors explore a range of different characteristics specific to the three methods explored as outcomes (implants, injectables and pills), which builds substantially on a currently limited literature base. Clarification of the distinct study samples, derived from the cohort study, would improve the methods section; use of a study selection flowchart is recommended. As described in the feedback for the methods section, it is unclear why the authors used follow-up data from different survey rounds when comparing Nairobi and Homa Bay, thereby inducing systematically different follow-up times between the samples. Given the authors’ inclusion of women who were pregnant at enrollment (more than one-quarter of non-users at baseline in both samples), I expected to see some assessment of or adjustment for postpartum status, as this is likely to shape how women act on beliefs to make decisions about contraception. Other specific suggestions, which I hope will improve the clarity and contribution of this important work, are itemized by section. A few grammatical errors, verb tenses, and missing words are also present throughout the manuscript and require attention before finalization.

Response: We would like to thank the reviewer for the comments and feedback. We concur with the reviewer's observations and have taken into consideration most of the suggestions. More specifically, we have revised the analysis by aligning the study sample for the two sites. The revised analysis is based on the 12-month follow-up for the two sites. We have also considered including a flow diagram to make it easier for readers to understand how we derived the analytical sample. We have revised Table 1 to present characteristics on the analytical sample (women who adopted an injectable, pill or implant by follow-up) compared with the characteristics of women who adopted other methods (including female sterilization, IUD, Condom, Lactational Amenorrhea Method, Rhythm and withdrawal methods) as well as women who did not adopt any method at follow up. The analysis is restricted throughout to only women who were not using any method or pregnant at baseline but were aware of pills, injectables and implants at the time of baseline survey. We hope this will be easier for readers to follow. We checked the possibility that women with a recent birth and thus likely to be breastfeeding and amenorrheic at follow-up might make different method choices from other women by comparing those who were pregnant at baseline with non-pregnant women but found no significant difference in the multivariate model. Responses to other questions/comments can be found below. Lastly, the manuscript has been reviewed and edited by a native English speaker who is also the fifth (last) author.

ABSTRACT

• On line 29, the authors use “round 2” to talk about follow-up data collected as part of the mentioned longitudinal study. This terminology is confusing because the study design has not yet been described. Please review the abstract accordingly.

Response: We have revised the text to make it clear [LN 22-27]. W have also revised the result section taking into consideration results using only 12-month follow-up data for both sites [LN 28-40]

INTRODUCTION

• On line 54, the authors state: “Fear of side effects, damage to health, menstrual disruption, and long-term infertility have been widely documented, including for Kenya [6-8].” Sentence should be clarified to state that these are the barriers mentioned in the previous sentence.

Response: We have revised as recommended by the reviewer [LN 53-53]

MATERIALS AND METHODS

• On line 94, please define acronym “NUHDSS” at first use. 

Response: the acronym “NUHDSS” is already defined in the preceding section on study site [LN74].

• Lines 103-110 describe how the authors arrived at the analytic sample. A figure would be helpful here, particularly since a broader sample is included in Tables 1-2, compared to the final analytic sample for the specific contraceptive outcomes in Tables 3-5. This would be especially helpful given the difference in follow-up times between Nairobi and Homa Bay samples.

Response: We would like to thank the reviewer for this suggestion. To describe how we derived our analytical sample, we have included a flowchart diagram indicating method choice at 12-month follow-up among women not using any method at baseline but were aware of pills, injectables and implants at the time of the survey [Figure 1].

• On lines 103-106, the authors describe the cohort study samples from the two sites. While there was no “round 3” conducted in Nairobi, why did the authors choose to use the different follow-up rounds for the two sites? Why not just restrict them both to round 2, given that the follow-up time between the baseline and follow-up samples would be more comparable? In the methods description, it does seem like round 2 data is available in Homa Bay, so this should be used instead of round 3. If not adjusting for time since baseline belief in each model, both models should at least include comparable samples in terms of follow-up time.

Response: We have analyzed round 2 data for the rural site to make it comparable with the urban site. We certainly lose much statistical power by using round 2 rather than round 3 in Homa Bay but there is only one serious difference in AORs, namely with safe for long time use which becomes insignificant. (LN 136-138, Figure 1]

• On line 112, the authors state, “…1866 women during the third round (91% of those interviewed in round 2).” This should be stated in terms of the retention from the original sample, not the round 2 sample. This recommendation can be ignored if the sample for Homa Bay is revised to include round 2 as follow-up only.

Response: We changed 91% to 86 % to state the retention from the original sample. [LN 110]

• On line 125-126, the authors state, “Note that beliefs b and c are valid while beliefs a, d, 

and e are erroneous.” Why do the authors make the distinction between beliefs about contraception that are (in their mind) “valid” and others are considered “erroneous”? If these beliefs are shaped by women’s social networks, communities, etc. and, in turn, inform women’s contraceptive decision-making, then why is this distinction important?

Response: Thank you for this observation. We concur that women’s beliefs about contraceptives are valid since it is in their minds. That said, we distinguish which beliefs are erroneous from an analytical perspective because this would have an implication on the education/counseling programme. There is need to demystify erroneous women’s beliefs such as ‘contraceptives cause infertility to improve uptake as such misconception could be attributed to incorrect knowledge, distortions of truths, or dissemination of false information. 

Additionally, the authors should include some mention of the large body of literature which ties contraceptive use/non-use to beliefs about contraception and infertility. 

Response: We have included some published literature [LN 125]

• On line 126, the authors mention, “Because of evidence on the importance of social influences…” Please cite relevant literature here or in the introduction.

Response: We have cited published literature [LN 125]

• On lines 130-131, the authors mention that, “Women's perception of their partners' approval of the method was also ascertained; in the analysis, don't know responses were combined with disapproval.” Why were “don’t know” responses combined with disapproval and not approval? For what proportion of the sample did this affect? I am particularly interested in this given the significant findings discussed later in the manuscript, which show a strong influence of perceived partner disapproval and contraceptive choice in Nairobi. Was there a difference between the proportion of women who selected “don’t know” between urban and rural sites that may have related to the observed differences?

Response: The decision to combine ‘don’t know’ responses’ with disapproval was the authors’ discretion after careful evaluation of the difference it would make on the results. Most past users of the methods who gave ‘don’t know responses’ tended to be dissatisfied with the method.

• On line 146, the word “variable” is meant to be plural (i.e., variables).

Response: Changed [LN 147]

• Participants who were pregnant at baseline were included in the analysis, accounting for more than one-quarter of all non-users at baseline. Was there any assessment of if method choice was shaped by whether or not women were postpartum? I expected to see some assessment of or adjustment for postpartum status, as this is likely to shape how women act on beliefs to make decisions about contraception. I understand that sample sizes are certainly an issue, but feel strongly that postpartum status be recognized in this analysis.

Response: We checked the possibility that women with a recent birth and thus likely to be breastfeeding and amenorrheic at follow-up might make different method choices from other women by comparing those who were pregnant at baseline with non-pregnant women but found no significant difference in the multivariate model

RESULTS

• On line 178-179, the authors state: “In Nairobi, 74% were using a method of contraception at baseline compared with 65% in Homa Bay.” In the methods section, the analytic sample is described to be women who were not using contraception at baseline, given that the focus of the analysis is contraceptive method adoption between baseline and follow-up. It is confusing to have Tables 1-2 present data for the full sample of women from the existing cohort, as many of them are not eligible for participation in the present study since they are not “at risk” for the outcome.

Response: We have removed the table with the full sample. Our initial intention was to show if the socio-demographic characteristics of non-users differed with current users. Given that the study if focused on method choice at follow-up among nonusers at baseline, the comparison does not add any value to the study. We have presented a Table that shows characteristics of married or cohabiting women aged 15–39 years by method choice at follow-up among non-users at baseline. We have categorized the method choice into three groups: 1) adopted implants, injectables or pills; 2 adopted other methods (include female sterilization, IUD, Condom, Lactational Amenorrhea Method, Rhythm and withdrawal methods); and 3) Non-adaptors, that is, non-users at baseline who did not adopt any method at 12-month follow-up [See Table 2, LN 192]

• If Tables 3-5 present results for the subsample of women who were not using at baseline, but had adopted an injectable, pill or implant by follow-up, Table 1 should reflect these same populations for comparison. For example, if the authors feel it is important to retain information on the full cohort in this paper, Table 1 might show three categories in each site: contraceptive users at baseline vs. non-users at baseline who do not adopt implants, injectables or pills vs. non-users at baseline who adopt implants, injectables or pills (i.e., analytic sample for Tables 3-5, main analyses). This would make it easier, from a reader’s perspective, to compare the population of women in the analytic sample to other women surveyed.

Response: As we have indicated above, we revised Table 1 to present characteristics on the analytical sample (women who adopted an injectable, pill or implant by follow-up). We have compared the characteristics of the analytical sample with those of women who adopted other methods (including female sterilization, IUD, Condom, Lactational Amenorrhea Method, Rhythm and withdrawal methods) as well as women who did not adopt any method at follow up. The analysis is restricted throughout to only women who were not using any method or pregnant at baseline but were aware of pills, injectables and implants at the time of baseline survey. We hope this will be easier for readers to follow. 

• With the previously suggested revision, update lines 176-186 to provide comparison between the analytic sample and the other two samples derived from the same cohort.

Response: We have revised the text as recommended by the reviewer. [LN 178-191]

• In Table 3, please define “ns” in the notes under the table as “non-significant”.

Response: We have inserted actual p-values instead of using an asterisk. We have changed Table 3 to Table 2. 

• Tables 4-5 should include Ns to specify analytic sample sizes.

Response: We have revised the Ns as recommended by the reviewer. We have changed Tables 4-5 to Tables 3-4. 

• Clarify throughout the results—and in other sections—when “husband’s approval” is stated that this is “perceived husband approval”, as this measure is reported by the woman herself (e.g., lines 240, 260, etc.)

Response: We have revised the as recommended by the reviewer. 

• Line 276 includes a typo for the word “above”

Response: We have delete the word ‘above’ [LN 278]

DISCUSSION

• The paragraph articulated in lines 309-317, connection between prior contraceptive use and future use relating to unmet need and programs, is excellent. Bravo!

Response: We are grateful for the compliment

• Starting on line 329, the authors focus the discussion on the contrast observed in the relationship between contraceptive beliefs and method-specific use across the two sites. See earlier comment in methods section about making these two samples more comparable with the use of round 2 data from Homa Bay to reflect the same follow-up times between baseline and follow-up for the two samples.

Response: As we already stated earlier, we have analyzed round 2 data for both rural and urban sites to make it comparable. We certainly lose much statistical power by using round 2 rather than round 3 in Homa Bay but there is only one serious difference in AORs, namely with safe for long time use which becomes insignificant. The effect of the rest of the attributes remains the same though with reduces magnitudes. Certainly, this may be due to time factors, however, it is beyond the capacity of the current study to prove. It is also possible that beliefs changed between baseline and 12-month follow-up, perhaps because of an unsatisfactory experience of a social network member.

• Starting on line 336, the authors describe the perplexing association between perceived husband’s approval and method choice, comparing the two sites. See earlier comment in methods section about the proportion of women who responded “don’t know” and how they were reclassified as having partners that “disapproved” of the method. Given these inconsistent results across the two sites, I remain curious about the proportion of women in each site whose responses were reclassified and how this reclassification might have misclassified women with uncertainty about their partners’ beliefs as having 

“disapproving” or hostile partners and therefore shaped the observed results.

Response: We recognize that women in Homa Bay were more likely to give ‘don’t know’ responses than those in Nairobi (about 10% versus 1%) when asked about perceived husband approval of the method. Although ‘don’t know’ responses were reclassified as disapproval, this is unlikely to account for the contrast—most past users who gave ‘don’t know’ responses were also dissatisfied with the method. [LN 343-347]

• On lines 344-346, the authors note that, “Beliefs about menstrual disturbance and safety for long-term use were related to method choice in the rural sample but not in Nairobi”. Perhaps women in Homa Bay value menstrual regularities more than women in Nairobi, seeing menstrual disturbances as a reflection of poor health, potential challenge in their relationships, potential indicator of later infertility, etc. whereas women in Nairobi conceptualize menstrual disturbances differently (e.g., that no bleeding is good, not a sign of poor health). See Polis 2018, ‘“There might be blood”: Scoping review of women’s responses to contraceptive-induced menstrual bleeding changes’ and other articles focused on how changes in menstruation affect women differently.

• 

Response: After comparing the results for the two sites using round 2 data, we found a lack of importance of the beliefs about menstrual disturbance and safety for long-term use on method choice in both sites. This contrasts with the literature noting that menstrual irregularities as an important factor in method choice. We agree with the reviewer that women do value menstrual regularities, that is, seeing menstrual disturbances as a reflection of poor health, a potential challenge in their relationships, and a potential indicator of later infertility. [LN 353-361]

• On lines 346-348, the authors highlight how the sites are distinct, indicating that their comparison might not be appropriate. Given the focus on the site-specific differences throughout the results and flagged in the abstract, this statement should come earlier in the article. Authors should clearly indicate that the two sites were selected due to the availability of data in the cohort study, etc. and not due to their comparability overall. 

Response: We have indicated under study settings that although the two study populations are in the same country, they differ radically in ethnicity, education, and other respects such as socioeconomic activities. [LN 83]

• I appreciate the authors’ focus on the negative results in lines 349-361. This is an important contribution—equally as valuable as the positive results, as the authors note!

Response: We are grateful for the compliment

• As flagged in the methods, the authors should reference literature on social networks and contraception to contextualize the results outlined in lined 358-361.

Response: We have inserted two relevant literatures [LN 372]

o On line 367-368, the authors state that women’s contraceptive beliefs may have changed “perhaps because of an unsatisfactory experience of a social network member”, which affected the insignificant relationships between many beliefs and later use of methods. Given the last point about the lack of observed association between social network experiences and women’s contraceptive choices, I don’t think this makes sense as a plausible explanation.

Response: We have replaced the sentence “unsatisfactory experience of a social network’’ with “counselling by a health provider”. [LN 383]

CONCLUSION

• On lines 383-384, the authors end with “Women's choices about which method to use should be based, as far as possible, on correct knowledge.” This seems like a bit of a leap. The focus should not be on women’s “correct knowledge” but instead on how valid concerns, misconceptions, fears, etc. of contraceptive attributes require attention from the field.

Response: We deleted the last sentence “Women's choices about which method to use should be based, as far as possible, on correct knowledge.”LN387-395]

REVIEWER #2

The paper is well written. It adds into existing literature on contraceptive method beliefs that act as facilitators or barriers to contraceptive uptake and continuation with use.

Introduction:

• The last statement on strengths should be moved to the discussion section (line 74-76)

Response: Thank you for this recommendation. We have moved the statement on the strength to the discussion section. [LN 376-378]

Methods

• Lines 93/100-Provide a rationale for limiting the eligibility criteria to women aged 15-39 years? Why not women of reproductive age,15-49years?

Response: We have clarified that only married or cohabiting women aged 15-39 years at the time of recruitment were eligible to participate. The restriction on the upper age limit was deliberate to allow follow-up of women when they were more likely to be at risk of pregnancy compared to unmarried, non-cohabiting or older women. [LN 99-101]

• Line 118: How were the 11 attributes selected? The five items related to health concerns and safety have some overlap. For instance, unpleasant side effects might also include option c (disruption of regular menses). Was the tool/questions pretested before administration?

Response: We reviewed existing literature and more than 30 instruments on fertility preferences and reasons for non-use of contraception conducted in low-, middle-, and high-income countries, and compiled question items by themes. This includes instruments from the DHS, the Determinants of Unintended Pregnancy Risk study in New Orleans, the US-based National Survey of Family Growth, and the Fog Zone study by the Guttmacher Institute. Subsequently, a new questionnaire was developed using the compilation of the question items. A draft instrument was reviewed through a consultative process with dozens of experts in the field.

Results

• line 198-199 Was the use of condoms especially in Homa Bay mainly for contraceptives or also for dual protection?

Response: 

Participants were asked the current method they were currently using. if more than one method was mentioned, then the highest method in the list was selected. Condoms are used as for dual protection-against HIV/STI and pregnancy. Homa Bay being a high HIV risk area is true that many couples may use condoms for dual protection. Our data attest to this, condoms were adopted more frequently in Homa Bay than in Nairobi.

• Were there any differences in the baseline characteristics of the women in the two counties?

Response: We have indicated in Table how the characteristics of the two study populations differed in terms of socio-demographic background-age, level of education and fertility preference. We have also provided in the Methods section how the two settings are different. [LN 78-87]

• Was data collected on discontinuation or method switching between round 2 and round 3 for Homa Bay?

Response: Yes, the study that provided data for this analysis collected data on discontinuation or method switching between rounds. However, the current analysis is focused on examining which method-specific beliefs influence subsequent adoption of particular hormonal methods in urban and rural Kenya. We also consider the influence of past experience with specific methods.

Discussion

• Lines 290/1-Were the women asked whether the method they chose was their preferred option?

Response: One section of the questionnaire collected information on women’s perceptions concerning method attributes. These were asked about eight methods but here we present data only for injectables, pills and implants, a restriction dictated by the fact that insufficient numbers adopted other methods to sustain analysis in both sites. All women who had heard of the specific method were asked their perceptions of eleven method attributes, regardless of whether or not they had ever used the method. Women were first asked whether the method was easy to obtain and whether it was, in their opinion, easy to use. Perceived effectiveness was ascertained by asking respondents whether or not they considered the method to be “very effective at preventing pregnancy”. Five items related to health concerns and safety. Specifically, women were asked whether they thought the method was likely to cause: (a) serious health problems; (b) unpleasant side effects; (c) disruption to regular menses; (d) long term infertility; (e) dangers if used for a long time without taking a break. 

---

## [Decision Letter · Decision Letter 1]

27 May 2021

Method-specific beliefs and subsequent contraceptive method choice: results from a longitudinal study in urban and rural Kenya

PONE-D-20-34145R1

Dear Dr. Odwe,

We’re pleased to inform you that your manuscript has been judged scientifically suitable for publication and will be formally accepted for publication once it meets all outstanding technical requirements.

Kind regards,

Philip Anglewicz, PhD

Academic Editor

PLOS ONE

Additional Editor Comments (optional):

Reviewers' comments:

Reviewer's Responses to Questions

**Comments to the Author**

1. If the authors have adequately addressed your comments raised in a previous round of review and you feel that this manuscript is now acceptable for publication, you may indicate that here to bypass the “Comments to the Author” section, enter your conflict of interest statement in the “Confidential to Editor” section, and submit your "Accept" recommendation.

Reviewer #1: All comments have been addressed

Reviewer #2: All comments have been addressed

2. Is the manuscript technically sound, and do the data support the conclusions?

Reviewer #1: Yes

Reviewer #2: Yes

3. Has the statistical analysis been performed appropriately and rigorously? 

Reviewer #1: Yes

Reviewer #2: Yes

4. Have the authors made all data underlying the findings in their manuscript fully available?

Reviewer #1: Yes

Reviewer #2: Yes

5. Is the manuscript presented in an intelligible fashion and written in standard English?

Reviewer #1: Yes

Reviewer #2: Yes

6. Review Comments to the Author

Reviewer #1: Thank you for revising the manuscript according to the feedback from reviewers. The revised manuscript is much clearer and easier to follow, especially with the revised description of the analytic samples. The discussion and conclusions are more focused too. Well done.

Reviewer #2: (No Response)

7. PLOS authors have the option to publish the peer review history of their article (what does this mean?). If published, this will include your full peer review and any attached files.

Reviewer #1: No

Reviewer #2: No

---

## [Editor Report · Acceptance letter]

11 Jun 2021

PONE-D-20-34145R1 

Method-specific beliefs and subsequent contraceptive method choice: results from a longitudinal study in urban and rural Kenya 

Dear Dr. Odwe:

I'm pleased to inform you that your manuscript has been deemed suitable for publication in PLOS ONE. Congratulations! Your manuscript is now with our production department. 

Kind regards, 

on behalf of

Associate Professor Philip Anglewicz 

Academic Editor

PLOS ONE